# Learning Safe Policies with Cost-sensitive Advantage Estimation

## Abstract

Reinforcement Learning (RL) with safety guarantee is critical for agents performing tasks in risky environments. Recent safe RL algorithms, developed based on Constrained Markov Decision Process (CMDP), mostly take the safety requirement as additional constraints when learning to maximize the return. However, they usually make unnecessary compromises in return for safety and only learn sub-optimal policies, due to the inability of differentiating safe and unsafe state-actions with high rewards. To address this, we propose Cost-sensitive Advantage Estimation (CSAE), which is simple to deploy for policy optimization and effective for guiding the agents to avoid unsafe state-actions by penalizing their advantage value properly. Moreover, for stronger safety guarantees, we develop a Worst-case Constrained Markov Decision Process (WCMDP) method to augment CMDP by constraining the worst-case safety cost instead of the average one. With CSAE and WCMDP, we develop new safe RL algorithms with theoretical justifications on their benefits for safety and performance of the obtained policies. Extensive experiments clearly demonstrate the superiority of our algorithms in learning *safer and better* agents under multiple settings.

## 1 Introduction

In recent years, Reinforcement Learning (RL) has achieved remarkable success in learning skillful AI agents in various applications ranging from robot locomotion (Schulman et al., 2015a; Duan et al., 2016; Schulman et al., 2015c), video games (Mnih et al., 2015) and the game of Go (Silver et al., 2016; 2017). These agents are either trained in simulation or in risk-free environments, and the deployed RL algorithms can focus on maximizing the cumulative return by exploring the environment arbitrarily. However, this is barely workable for real-world RL problems where the safety of the agent is important. For example, a navigating robot cannot take the action of crashing into a front obstacle even if the potential return on reaching the target faster is higher. Actually, in reality, some states or actions might be unsafe and harmful to the system, and the agent should learn to avoid them in deployment when performing certain tasks. Conventional RL algorithms do not particularly consider such safety-constrained environments, which limits their practical application.

Recently, *Safe Reinforcement Learning* (Garcıa & Fernández, 2015; Mihatsch & Neuneier, 2002; Altman, 1999) has been proposed and drawn increasing attention. Existing safe RL algorithms generally fall into two categories based on whether or not the agents are required to always stay safe during learning and exploration. The algorithms with *exploration safety* (Dalal et al., 2018; Pecka & Svoboda, 2014) insist that safety constraints never be violated even during learning, and thus they usually require certain prior knowledge of the environment to be available, *e.g.*, in the form of human demonstrations. Comparatively, *deployment safety* (Achiam et al., 2017; Chow et al., 2018) RL algorithms train the agents from interaction with the environment and allow safety constraints violation during learning to some extent. This is reasonable since whether a state is safe will not be clear until the agent visits that state. Since human demonstrations are too difficult or expensive to collect in some cases and may not cover the whole state space, we focus on deployment safety in this work.

RL problems with deployment safety are typically formulated as Constrained Markov Decision Process (CMDP) (Altman, 1999) that extends MDP by requiring the agent to satisfy cumulative cost constraints in expectation in the meanwhile of maximizing the expected return. Leveraging the

success of recent deep learning powered policy optimization methods (Schulman et al., 2015b), Constrained Policy Optimization (CPO) (Achiam et al., 2017) makes the first attempt on high-dimensional control tasks in continuous CMDPs. However, CPO only considers the total cost of a trajectory of a sequence of state-action pairs during policy optimization. It does not differentiate the safe state-action pairs from the unsafe ones in the trajectories. Due to such incapability of exploiting the intrinsic structure of environments and trajectories, CPO sacrifices too much on the expected return for learning the safety policy.

In this work, we propose *Cost-sensitive Advantage Estimation* (CSAE) which generalizes the conventional advantage estimation for safe RL problems by differentiating safe and unsafe states, based on the cost information returned by the environment during training. CSAE depresses the advantage value of unsafe state-action pairs but controls effects upon their adjacent safe state-actions in the trajectories. Thus, the learned policy can maximally gain rewards from the safe states. Based on CSAE, we develop a new safe RL algorithm with proved monotonic policy performance improvement in terms of *both* safety and return from safe states, showing superiority over other safe RL algorithms. Moreover, to further enhance the agent's ability of enforcing safety constraints, we propose *Worst-case Constrained Markov Decision Process* (WCMDP), an extension of CMDP by constraining the cumulative cost in worst cases through the Conditional Value-at-Risk (Tamar et al., 2015), instead of that in expectation. This augmentation makes the learned policy not only safer but also better, both experimentally and theoretically.

With CSAE and WCMDP, we develop a new safe RL algorithm by relating them to trust region methods. We conduct extensive experiments to evaluate our algorithm on several constrained robot locomotion tasks based on Mujoco (Todorov et al., 2012), and compare it with well-established baselines. The results demonstrate that the agent trained by our algorithm can collect a higher reward, while satisfying the safety constraints with less cost.

## 2 RELATED WORK

Safe Reinforcement Learning has drawn growing attention. There are various definitions of 'safety' in RL (Garcıa & Fernández, 2015; Pecka & Svoboda, 2014), *e.g.*, the variance of return (Heger, 1994; Gaskett, 2003), fatal transitions (Hans et al., 2008) and unknown states (Garcıa et al., 2013). In this paper, we focus on the RL problems with trajectory-based safety cost, under the constrained MDP (CMDP) framework. Through Lagrangian method, Geibel & Wysotzki (2005) propose to convert CMDP into an unconstrained problem to maximize the expected return with a cost penalty. Though such a problem can be easily solved with well-designed RL algorithms, *e.g.* (Schulman et al., 2015b; 2017), the trade-off between return and cost is manually balanced with a fixed Lagrange multiplier, which cannot guarantee safety through learning. To address this, inspired by trust region methods (Schulman et al., 2015b), Constrained Policy Optimization (Achiam et al., 2017) (CPO) establishes linear approximation to the safety constraint and solves the corresponding optimization problem in the dual form. Compared with previous CMDP algorithms, CPO scales well to high-dimensional continuous state-action spaces. However, CPO does not distinguish the safe states from the unsafe ones in the training process, limiting its performance in the return.

Besides developing various optimization algorithms, some recent works also explore other approaches to enhance the safety constraints, e.g., adopting the Conditional Value-at-Risk (CVaR) of the cumulative cost as the safety constraint (Tamar et al., 2015). Along this direction, Tamar et al. (2015) develop a gradient estimator through sampling to optimize CVaR with gradient descent. Prashanth (2014) further applies this estimator to CVaR-Constrained MDP to solve the stochastic shortest path (SSP) problem.

Our work considers a similar framework to CPO (Achiam et al., 2017), but it treats states differently by extending Generalized Advantage Estimation (Schulman et al., 2015c) to be safety-sensitive. Our proposed CSAE can boost the policy performance in terms of the return while ensuring the safety property. Moreover, our algorithm with WCMDP is safer than CPO in terms of constraint violation ratio during learning.

There are also some non-CMDP based algorithms for safe RL that are not in the scope of this work. In (Dalal et al., 2018), a linear safety-signal model is built to estimate per-step cost from state-action pairs and rectify the action into a safe one. However, this method requires a pre-collected dataset

to fit the linear cost estimation model, which limits its application. Similarly, Cheng et al. (2019) augment the model-free controller to enforce safety per step by designing a modle-based controller with control barrier functions (CBFs). Some works introduce Lyapunov functions to build safe RL algorithms. For example, Berkenkamp et al. (2017) apply Lyapunov functions for safely recovering from exploratory actions, while Chow et al. (2018) construct Lyapunov functions that explicitly model constraints.

## 3 PRELIMINARIES

A standard Markov Decision Process (MDP) (Sutton et al., 1998) is defined with a tuple $(\mathcal{S}, \mathcal{A}, P, R, \gamma, \mu)$, where $\mathcal{S}$ and $\mathcal{A}$ denote the set of states and actions respectively, $P : \mathcal{S} \times \mathcal{A} \times \mathcal{S} \rightarrow [0, 1]$ is the transition dynamics modeling the probability of transferring from state $s$ to $s'$ after taking action $a$, $R(s, a, s')$ represents the reward function during this transition, $\gamma \in [0, 1]$ is the discount factor and $\mu : \mathcal{S} \mapsto [0, 1]$ denotes the starting state distribution.

An MDP agent is usually equipped with a policy $\pi(a|s)$, which denotes the probability distribution over actions $a$ given a state $s$. The performance of a policy $\pi$ is measured with the expected discounted total reward $J(\pi) = \mathbb{E}_{\tau \sim \pi, s_0 \sim \mu}[\sum_{t=0}^{\infty} \gamma^t R(s_t, a_t, s_{t+1})]$, where $\tau = (s_0, a_0, s_1, \dots)$ is a trajectory generated by following policy $\pi$. RL algorithms for MDPs try to find the policy $\pi^*$ that achieves the highest reward, *i.e.*, $\pi^* = \arg\max_\pi J(\pi)$. They commonly use the value function $V_\pi(s) = \mathbb{E}_{\tau \sim \pi}[\sum_{t=0}^{\infty} \gamma^t R(s_t, a_t, s_{t+1})|s_0 = s]$, the action value function $Q_\pi(s, a) = \mathbb{E}_{\tau \sim \pi}[\sum_{t=0}^{\infty} \gamma^t R(s_t, a_t, s_{t+1})|s_0 = s, a_0 = a]$ and the advantage function $A_\pi(s, a) = Q_\pi(s, a) - V_\pi(s)$. The discounted future state distribution will also be useful, which is defined as $d^\pi(s) = (1 - \gamma)\sum_{t=0}^{\infty} \gamma^t P(s_t = s|\pi)$.

Constrained Markov Decision Process (CMDP) (Altman, 1999) extends MDP to environments with safety cost that could harm the agent when undesired actions are taken. As various safety costs may exist in a single CMDP, we relate them with $m$ cost functions $\{C_1(s, a, s'), \dots, C_m(s, a, s')\}$, each of which denotes the cost an agent receives for each transition $(s, a, s')$ (similar to reward functions). Let $C_i(\tau) = \sum_{t=0}^{\infty} \gamma^t C_i(s_t, a_t, s_{t+1})$ denote the cumulative cost along a trajectory $\tau$ generated from policy $\pi$. We consider a trajectory-based cost constraint in CMDP, which limits the cumulative cost in expectation $J_{C_i} = \mathbb{E}_{\tau \sim \pi, s_0 \sim \mu}[C_i(\tau)]$ with value $d_i$. Then safe RL aims to learn the policy $\pi$ under CMDP by solving the following problem,

$$\pi^* = \arg\max J(\pi), \text{ s.t. } J_{C_i} = \mathbb{E}_{\tau \sim \pi, s_0 \sim \mu}[C_i(\tau)] \leq d_i, i = 1, \dots, m. \quad (1)$$

Safe RL algorithms search for the policy $\pi^*$ that achieves the maximal cumulative reward and meanwhile does not violate the imposed safety constraints on the costs. In the following, analogous to the definition of value functions (*i.e.*, $V_\pi$, $Q_\pi$ and $A_\pi$), we use $V_\pi^{C_i}$, $Q_\pi^{C_i}$ and $A_\pi^{C_i}$ to denote the cost-value functions *w.r.t.* cost function $C_i$.

## 4 METHOD

In this section, we develop a policy gradient based algorithm for solving the safe Reinforcement Learning problem in Equation 1. We will first derive a novel cost-sensitive advantage estimation method and present theoretical guarantees on the performance of its learned policy in terms of rewards from safe states. Then, we further develop a worst-case constrained MDP to augment the safety guarantee for learning policies. Finally, we present our safe RL algorithm in details.

### 4.1 COST-SENSITIVE ADVANTAGE ESTIMATION

Conventional policy optimization methods (either for RL or for Safe RL) usually model the policy with a parametric function approximator (*e.g.*, neural networks), and directly optimize the expected return $J(\pi_\theta)$, where $\pi_\theta$ denotes the policy parameterized with $\theta$. The gradient estimator $g$ for policy gradient methods (Schulman et al., 2015b;c) generally takes the following form:

$$g = \mathbb{E}\left[\sum_{t=0}^{\infty} \Phi(s_t, a_t)\nabla_\theta \pi_\theta(a_t|s_t)\right], \quad (2)$$

where $\Phi(s_t, a_t)$ is responsible for guiding the policy updating direction and one popular choice for $\Phi(s_t, a_t)$ is Generalized Advantage Estimator (GAE) (Schulman et al., 2015c) which substantially reduces the variance of policy gradient estimate. The formulation for GAE[1] is given by

$$\hat{A}_t^{\text{GAE}(\gamma,\lambda)} := \sum_{l=0}^{\infty} (\gamma\lambda)^l \delta_{t+l}, \tag{3}$$

where $\lambda \in [0, 1]$ is a hyper-parameter. When $\lambda = 0$, it reduces to one-step TD error estimator; when $\lambda = 1$, it reduces to the empirical return estimator.

**Cost-sensitive Advantage Estimation**    Existing safe RL algorithms directly deploy these estimators without adaptation to the specific feature of safe RL problems and fail to consider the safety requirement within the gradient estimation. For example, CPO (Achiam et al., 2017) uses environment reward to estimate the advantage function for policy optimization, without considering that some high-reward states may also be unsafe.

In safe RL, an unsafe state with high reward would bias policy update towards favoring such a state and wrongly encourage the agent to violate cost constraints, if directly applying the GAE estimator. A natural solution is to penalize the reward for unsafe states. However, it is difficult to adjust the penalty appropriately. Specifically, over-penalization would suppress visiting the nearby safe states with high reward as their $\Phi(s_t, a_t)$ will be negatively affected during bootstraping. On the other hand, the unsafe state cannot be avoided when the penalty is too small.

Since $\delta_t$ can be considered as an estimate of the advantage value of taking action $a_t$ at step $t$, the policy gradient estimator $g$ points to the direction of increasing $\pi(a_t|s_t)$ only if the advantage of $a_t$ is greater than zero. Therefore, to guarantee that agents can gain rewards mainly from safe states, we propose to generalize GAE for safe RL by zeroing the TD error $\delta$ of unsafe states to avoid the agents from further exploring these regions. This is given by

$$\hat{A}_t^{\text{CSAE}(\gamma,\lambda)} := \sum_{l=0}^{\infty} (\gamma\lambda)^l \alpha_{t+l} \delta_{t+l}, \tag{4}$$

where $\alpha_t$ is a binary variable denoting whether a transition $(s_t, a_t, s_{t+1})$ is safe ($\alpha_t = 1$) or not ($\alpha_t = 0$). Following standard assumption in safe RL (Achiam et al., 2017), given the returned cost from the environment in the training phase, $\alpha_t$ can be obtained by binarizing the cost value $C(s_t, a_t, s_{t+1})$, i.e., $\alpha_t = \mathbf{1}[C(s_t, a_t, s_{t+1}) > 0]$. With this new advantage estimation, the policy gradient estimator for safe RL is given by

$$g^{\text{CSAE}} = \mathbb{E}\left[\sum_{t=0}^{\infty} \hat{A}_t^{\text{CSAE}(\gamma,\lambda)} \nabla_\theta \pi_\theta(a_t|s_t)\right],$$

which is compatible with any policy gradient based methods.

**CSAE and Reward Reshaping**    The above CSAE is equivalent to a moderate reward reshaping to penalize the reward for unsafe states. More specifically, it replaces the reward value for an unsafe state with the expected one-step reward an agent can receive at this state:

$$\bar{R}(s_t, a_t, s_{t+1}) = \begin{cases} R(s_t, a_t, s_{t+1}), & \text{if } \alpha_t = 1, \\ \mathbb{E}_{a,s'\sim\tau}\left[R(s_t, a, s')\right], & \text{if } \alpha_t = 0. \end{cases} \tag{5}$$

Using this reshaped reward function induces the above CSAE advantage estimator. To see this, we use $r_t$ and $\bar{r}_t$ to substitute $R(s_t, a_t, s_{t+1})$ and $\bar{R}(s_t, a_t, s_{t+1})$, respectively, in the following and drop subscript $\pi$ from the value function for notation simplicity[2]. Following standard definition, at time step $t$, a $k$-step advantage estimation $A_t^{(k)}$ using the value function $V$ and our revised reward signal $\bar{r}$ can be expressed as

$$A_t^{(k)} = -V(s_t) + \bar{r}_t + \gamma\bar{r}_{t+1} + \cdots + \gamma^{k-1}\bar{r}_{t+k-1} + \gamma^k V(s_{t+k}). \tag{6}$$

---

[1]We use $\hat{A}_t^{\text{GAE}(\gamma,\lambda)}$ to denote $\hat{A}^{\text{GAE}(\gamma,\lambda)}(s_t, a_t)$.

[2]Note that the reward revision mechanism in Equation 5 is only used for advantage estimation. For fitting the value function during learning, we still use the original reward function $R(s, a, s')$.

By substituting one-step TD error $\delta_t$ and reward function (Equation 5) into Equation 6, the above advantage can be rewritten as

$$A_t^{(k)} = \sum_{l=0}^{k-1} \gamma^l \alpha_{t+l} \delta_{t+l}. \tag{7}$$

See the appendix for the complete proof. Analogous to GAE, CSAE can be obtained by taking the exponentially-weighted average of above $k$-step advantage: $\hat{A}_t^{\text{CSAE}(\gamma,\lambda)} := (1 - \lambda) \sum_{k=1}^{\infty} \lambda^{k-1} A_t^{(k)} = \sum_{l=0}^{\infty} (\gamma\lambda)^l \alpha_{t+l} \delta_{t+l}$. This provides another perspective, from reward reshaping, to interpret the proposed CSAE. As policy optimization methods will automatically force agents to find high-reward regions in the state space, using the averaged reward can prevent unsafe yet high-reward states from attracting the agent during learning.

From the reward reshaping perspective, another possible approach to deal with the cost is to include the cost $c_t$ in the reward by reshaping $r_t$ to $R_t = r_t + \lambda \times c_t$. But it is difficult to properly choose the trade-off parameter $\lambda$ due to: 1) if $\lambda$ has fixed value, it is not easy to balance $r_t$ and $c_t$ as their best trade-off varies across environments, as verified by Tessler et al. (2018). In contrast, our proposed method is free of hyperparameter tuning and easy to deploy. 2) if $\lambda$ is treated as the dual variable for safety hard constraints and updated in a similar way as PDO, the performance is worse than our method, due to the optimization difficulties, as justified in our experiments.

**Worst-Case Constraints** As discussed in Sec. 3, in a CMDP, the trajectory-based safety cost for cost function $C_i$ is computed and constrained in expectation, *i.e.*, $J_{C_i}(\pi) = \mathbb{E}_{\tau\sim\pi}[\sum_{t=0}^{\infty} \gamma^t C_i(s_t, a_t, s_{t+1})] \leq d_i$. However, this will certainly lead the agent to violate the constraints frequently during learning. To further enhance safety, we instead consider the worst cases and constrain the cost from the trajectories incurring largest cost.

We propose the Worst-case Constrained MDP (WCMDP), an MDP with a constraint on the CVaR of cost values (Tamar et al., 2015; Prashanth, 2014) in safe RL. It tries to find a policy that maximizes the cumulative return, while ensuring the conditional expectation of other cost functions given some confidence level $\beta$, to be bounded. Formally, for a cost function $C_i$ and a given $\beta \in (0, 1)$, the worst case constraint is given by

$$J_{C_i}^{\beta}(\pi) = \mathbb{E}_{\tau\sim\Delta_{\pi,\beta}} \left[ \sum_{t=0}^{\infty} \gamma^t C_i(s_t, a_t, s_{t+1}) \right], \tag{8}$$

where $\Delta_{\pi,\beta}$ is the set of top $\beta$ worst trajectories with the largest costs. We found the performance is robust to the value of $\beta$ and we empirically set $\beta = 0.1$. Accordingly, the safety constraint related to cost function $C_i$ is expressed as $J_{C_i}^{\beta}(\pi) \leq d_i$.

## 4.2 SAFE RL ALGORITHM WITH CSAE

Different from general RL problems, for safe RL, it is critical to ensure that the agent mostly gains reward from safe states and transitions. Thus, we are concerned with the following cost-sensitive return developed from the reshaped rewards in Equation 5 in safe RL:

$$J_{\text{safe}}(\pi) := \mathbb{E}_{\tau\sim\pi, s_0\sim\mu} \left[ \gamma^t \sum_{t=0}^{\infty} \bar{R}(s_t, a_t, s_{t+1}) \right], \tag{9}$$

where $\bar{R}(s_t, a_t, s_{t+1}) = \alpha_t R(s_t, a_t, s') + (1 - \alpha_t)\mathbb{E}_{a,s_{t+1}}[R(s_t, a, s')]$. Different from the conventional return that accumulates the rewards from both safe and unsafe states, the above reshaped return characterizes how much the agent can gain reward from safe state-actions. In this section, we demonstrate adopting the proposed CSAE in policy optimization would naturally optimize $J_{\text{safe}}$. To this end, we establish the following theoretical result that gives performance guarantees for the policies in terms of the cost-sensitive return $J_{\text{safe}}(\pi)$.

**Theorem 1.** *For any policies $\pi', \pi$ with $\epsilon^{\pi'} \doteq \max_s |\mathbb{E}_{a\sim\pi'}[\hat{A}_\pi^{\text{CSAE}(\gamma,\lambda)}(s,a)]|$, the following bound holds:*

$$J_{\text{safe}}(\pi') - J_{\text{safe}}(\pi) \geq \frac{1}{1-\gamma} \mathbb{E}_{\substack{s\sim d^\pi \\ a\sim\pi'}} \left[ \hat{A}_\pi^{\text{CSAE}(\gamma,\lambda)}(s,a) - \frac{2\gamma\epsilon^{\pi'}}{1-\gamma} D_{TV}(\pi'||\pi)[s] \right]. \tag{10}$$

Here $D_{TV}$ denotes the total variance divergence, which is defined as $D_{TV}(p||q) = \frac{1}{2}\sum_i |p_i - q_i|$ for discrete probability distributions $p$ and $q$. Due to space limit, we defer all the proofs to the appendix.

The above result bounds the difference of two policies in terms of the cost-sensitive return via the CSAE. Leveraging such a result, our safe RL algorithm updates the policy by

$$\pi_{k+1} = \arg\max_{\pi} \mathbb{E}_{s \sim d^{\pi_k}, a \sim \pi}[\hat{A}_{\pi_k}^{\text{CSAE}(\gamma,\lambda)}(s,a)] - \nu_k D_{TV}(\pi||\pi_k)[s]$$

$$\text{s.t.} \quad J_{C_i}^{\beta} = \mathbb{E}_{\tau \sim \Delta_{\pi,\beta}}[C_i(\tau)] \leq d_i, i = 1, \ldots, m. \tag{11}$$

In particular, from Equation 10, for appropriate coefficients $\nu_k$, the above update ensures monotonically non-decreasing return from safe states. Details of the practical implementation of this algorithm are provided in the appendix.

## 5 EXPERIMENTS

As this work targets at obtaining safer and better policies, through experiments we aim to investigate: 1) whether our designed CSAE is effective for guiding the policy optimization algorithm to achieve higher cumulative reward while satisfying safety constraints; 2) whether the new policy search algorithm induced from WCMDP can guarantee stronger safety without sacrificing the performance; and 3) whether our method is able to adjust the advantage value of each transition properly to better guide policy optimization. Therefore, we evaluate our methods on multiple high-dimensional control problems that mainly include two different tasks. 1) *Circle* (Schulman et al., 2015b) where the agent is required to walk in a circle to achieve the highest cumulative reward, but the safe region is restricted to lie in the middle of two vertical lines. 2) *Gather* where several apples are randomly placed in both safe and unsafe regions, and an agent should collect as many apples as possible from the safe regions and avoid entering the unsafe regions. In our experiments, the reward for collecting one apple is 10, and the cost is 1 for each time the agent walks into an unsafe region. See Fig. 3 for an example of the gather environment. For the circle environment, we use three different robot agents in Mujoco (Todorov et al., 2012), *i.e.*, point mass, ant and humanoid. For the gather environment, we conduct experiments with point mass and ant.

We use *CSAE* (Sec. 4.2) to denote the safe policy search algorithm equipped with our proposed cost-sensitive advantage estimation, and *CSAE-WC* to denote the algorithm that further includes worst-case constraints. We compare these two methods with three well-established baselines. *TRPO* (Schulman et al., 2015b): the most widely used policy optimization method; *CPO* (Achiam et al., 2017): the state-of-the-art safe RL algorithm for large-scale CMDP; *PDO*: a primal-dual optimization based safe RL algorithm (Achiam et al., 2017). For all the experiments, we use a multi-layer perceptron with two hidden layers of (64, 32) units as the policy network. Our implementation is based on rllab (Duan et al., 2016) and the Github repository[3]. The hyper-parameters for the environments and algorithms are given in the supplementary material.

**Results** The learning curves for all the methods and environments are plotted and compared in Fig. 1. The first row is the cumulative reward. As we are dealing with environments with safety cost, we only accumulate the rewards collected through safe transitions as an optimal safe RL algorithm should be able to acquire rewards from safe states and avoid high-reward unsafe states. We also visualize the full returns in Fig. 1 (second row) for completeness. From the results, one can observe that our *CSAE* surpasses CPO throughout all the environments. This demonstrates the effectiveness of CSAE for learning safe agents with higher rewards. Furthermore, with the help of worst-case constraints, *CSAE-WC* performs the best in terms of *rewards from safe states* for PointCircle and PointGather or comparably well for AntCircle, HumanCircle and AntGather, outperforming CPO. The second and third rows in Fig. 1 plot the cumulative cost and ratio of the safe trajectories[4] in all the trajectories at each sampling. Specifically, a safe ratio of 1 means all the collected trajectories are safe. From the results, the cost value of *TRPO* agents explodes as the training proceeds, while all the other three methods converges. Among them, *CSAE* achieves comparable cost value as *CPO* and higher safe ratio. CSAE-WC surpasses the other methods—it not only satisfies the constraint with less cost but also achieves highest safe ratio (nearly 1). These results clearly show that our method is effective at both enforcing safety and collecting more rewards, or it is *safer and better*.

---

[3]https://github.com/jachiam/cpo/
[4]One trajectory is counted as safe if its cumulative cost is smaller or equal to the constraint value $d$.

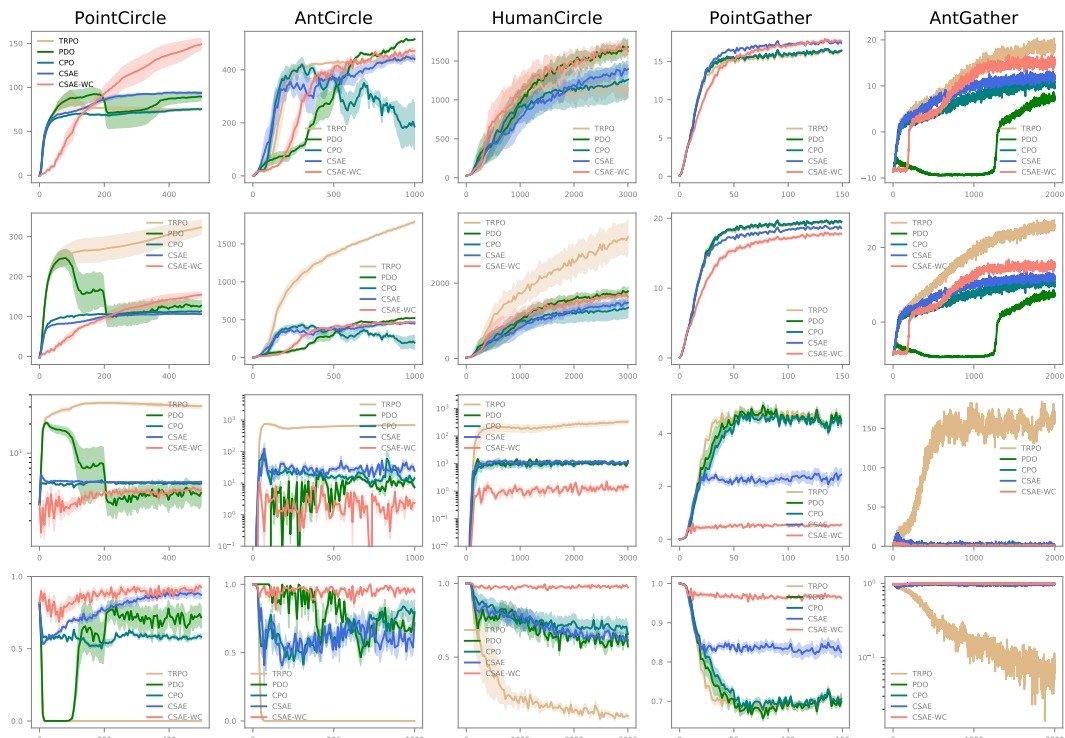

Figure 1: Learning curve comparison between our methods (CSAE and CSAE-WC) and the state-of-the-arts (TRPO, PDO, CPO) for five safe RL problems. First row: safe cumulative reward. Second row: total cumulative reward. Third row: cumulative cost. Fourth row: ratio of safe trajectories. $x$ axes denote the training iteration. (Best viewed in color). Each curve is obtained by averaging over five random runs. The standard deviation of different runs is visualized with the shade.

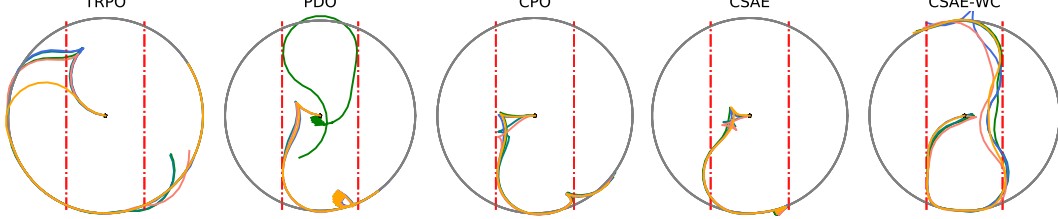

Figure 2: **Agents trained in PointCircle.** The grey circle denotes the path with highest reward. The two red dotted lines are the boundaries and the agent is constrained to run between them. Lines with different colors starting from the center are agent trajectories learned with different random seeds.

**Visualization** To intuitively justify our method indeed learns agents that take safer and better actions, we visualize agent trajectories for the circle task (Fig. 2) and the gather task (Fig. 3). Fig. 2 shows TRPO agent follows the circle specified by the reward function without considering constraints. The other safe RL agents can learn to obey the constraints to some extent. However, they do not perform well as they usually get stuck in a corner (*e.g.*, for PDO and CPO). Our CSAE-WC agents, however, can walk along the arcs and safe boundaries. Similar observations can be made in AntGather, where TRPO agent inevitably violates the constraint and rushes into unsafe regions (*i.e.*, the red squares). The other agents learn to avoid such cost but sacrifice the rewards. However, CSAE and CSAE-WC can work better to collect more rewards than others. In summary, both visualizations in Fig. 2 and Fig. 3 demonstrate the effectiveness of our method for learning better agents that generate more reasonable and safer trajectories.

**Analysis** We here investigate how our proposed CSAE helps the training process and the resulted agents. We use PointCircle as the environment to conduct the following analysis. First, we justify

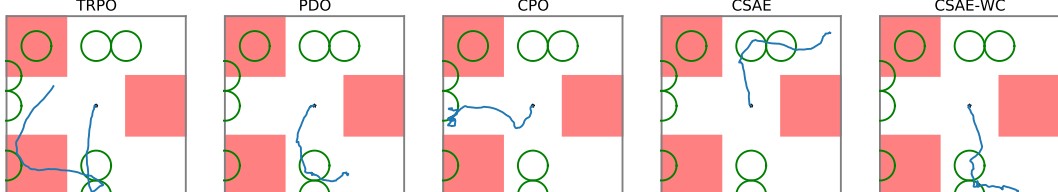

Figure 3: **Agents trained in AntGather.** The green circles denote the randomly placed apples to collect and red-colored squares are the unsafe regions. The blue lines are trajectories of an agent trying to explore the environment to collect apples.

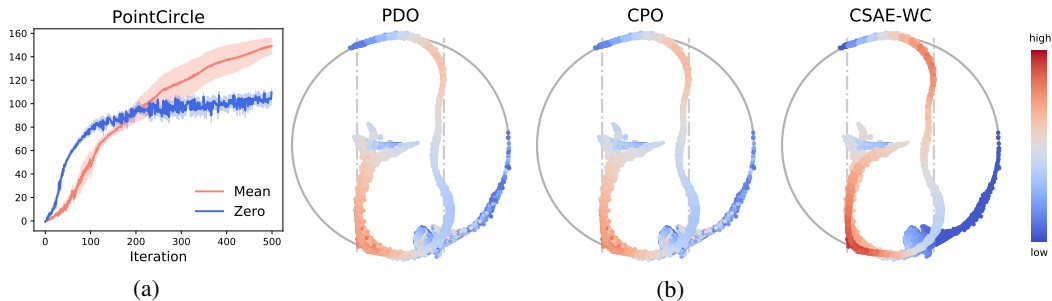

Figure 4: (a) Comparison of **average return on PointCircle** for different reward modifications. "Mean" is our method in Eqn. equation 5 and "Zero" reshapes reward into $\bar{r}_t = \alpha_t \times r_t$. (b) **Advantage value visualization**. Each colored dot represents a transition in the trajectories, whose intensity denotes relative value of the corresponding advantage. (Best viewed in color).

the method of replacing the reward with the expected one-step reward (Equation 5) for unsafe states. We compare it with a simple reward reshaping method that zeros the reward of unsafe transitions and plot their learning curves (of average return) in Fig. 4a. The results show that our method (denoted by "Mean" in Fig. 4a) performs much better. This indicates that our method can overcome the shortcomings of penalizing the reward of unsafe transitions not properly.

Second, it is important for safe RL algorithms to help the agent distinguish high-reward but unsafe states from the safe ones. To investigate the differences of safe RL algorithms (PDO, CPO and our CSAE-WC) in this ability, we sample 300 trajectories (100 from each method). For different algorithms, we use their deployed reward and value functions to estimate the advantage value for each transition in these trajectories. The advantage values are visualized in Fig. 4b, where more reddish means higher relative advantage value and bluish means lower value. From such visualization, one can observe that these three methods can recognize high-reward and safe state-actions by assigning higher advantage values, as shown in the left-bottom and right-top in Fig. 4b. However, our algorithm CSAE-WC prefers these safe and high-reward regions more with higher advantage values. Importantly, as shown in the right-bottom (unsafe but high-reward regions), our method gives state-actions within such regions much lower advantage. In contrast, PDO and CPO even assign above-the-average advantages to them. This result clearly demonstrates the superior and desired ability of our method to distinguish unsafe states from the safe ones for policy learning.

## 6 CONCLUSION

In this paper we consider Safe Reinforcement Learning and propose a novel CSAE method to appropriately estimate the advantage value for policy optimization under risky environments. Compared to conventional advantage estimation, CSAE eliminates the negative effect of high-reward but unsafe state-actions by depressing their advantages. To further enforce safety constraints, we augment the CMDP with the worst-case cost constraint and proposed WCMDP. We theoretically analyze their performance and safety benefits. We then develop a new safe RL algorithm which is shown effective for learning safer and better agents in multiple large-scale continuous control environments.

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

## 7    APPENDIX

### 7.1    POLICY OPTIMIZATION WITH WORST-CASE CONSTRAINTS

Since solving the exactly optimal policy is intractable for large-scale problems, policy gradient based methods represent the policy with a $\theta$-parameterized model, and try to search for the best policy within the parameter space $\Pi_\theta$, *i.e.*, $\pi^* = \arg\max_{\pi \in \Pi_\theta} J(\pi_\theta)$. Similarly, in the optimization problem induced from the worst case constrained policy optimization, we additionally require the policy to satisfy a set of safety constraints $\Pi_C^\beta$. In other words, we are optimizing the policy to achieve highest cumulative reward over the intersection of $\Pi_\theta$ and $\Pi_C^\beta$, which is formulated as

$$\max_{\pi \in \Pi_\theta} \quad J(\pi), \quad \text{s.t.} \quad J_{C_i}^\beta(\pi) \le d_i, \quad i = 1, \ldots, m.$$

Compared to CMDP objective in Equation 1, our proposed method requires the worst $\beta$-quantile trajectories (instead of the average cost) to still satisfy the safety constraints. This will yield a safer policy as proved later. Before presenting our algorithm in full details, the following result is given that is useful for connecting the worst case safety cost of two different policies and their difference.

**Theorem 2.** *Let $P_\beta$ denote the state transition probability $P$ of $\beta$-worst case trajectories. For any policies $\pi$ and $\pi'$, define $\epsilon_{C_i}^{\pi'} \doteq \max_s |\mathbb{E}_{a \sim \pi', s' \sim P_\beta}[C_i(s, a, s') + \gamma V_{C_i}(s') - V_{C_i}(s)]|$. Let $d_\beta^\pi$ denote the discounted future state distribution for the $\beta$-worst trajectories. Then the following bound holds:*

$$J_{C_i}^\beta(\pi') \le J_{C_i}^\beta(\pi) - \frac{1}{1-\gamma} \mathbb{E}_{\substack{s \sim d_\beta^\pi \\ a \sim \pi'}} \left[ A_\pi^{C_i}(s, a) + \frac{2\gamma \epsilon_{C_i}^{\pi'}}{1-\gamma} D_{TV}(\pi' || \pi)[s] \right]. \tag{12}$$

The above gives an upper bound on the worst case cost for policy $\pi'$. Explicitly constraining such an upper bound during the policy learning would enforce less cost constraint violation. Compared with the risk-sensitive CVaR models (Chow & Ghavamzadeh, 2014), this work is among the first to introduce such worst-case constraints into safe RL problems. Besides, it is also the first to present theoretical analysis on the expected worst-case cost of two policies, which is of independent interest.

We now show how to develop a practical algorithm for safe RL based on WCMDP and CSAE. Inspired by trust region methods (Schulman et al., 2015b), by replacing $J_{C_i}$ with $J_{C_i}^\beta$ and applying Theorem 2, we reformulate the update in Equation 11 into

$$\pi_{k+1} = \arg\max_\pi \mathbb{E}_{s \sim d^{\pi_k}, a \sim \pi}[\hat{A}_{\pi_k}^{\text{CSAE}}(s, a)]$$

$$\text{s.t.} \quad J_{C_i}^\beta(\pi_k) + \frac{1}{1-\gamma} \mathbb{E}_{\substack{s \sim d^{\xi \pi_k} \\ a \sim \pi}} \left[ A_{\pi_k}^{C_i}(s, a) \right] \le d_i, \tag{13}$$

$$D_{KL}(\pi || \pi_k) \le \delta, \quad i = 1, \ldots, m,$$

which is guaranteed to produce policies with monotonically non-decreasing returns from safe state-actions. Meanwhile, the policies will satisfy the original safety cost constraints.

### 7.2    ALGORITHM DETAILS

To efficiently solve Equation 13, we linearize the objective and cost constraint around $\pi_k$ and expand the trust region constraint to second order, similar to (Schulman et al., 2015b; Achiam et al., 2017). We use $\theta_k$ to denote parameters of policy $\pi_k$. Denote the gradient of objective and constraint for $J_{C_i}^\beta$ as $g$ and $b_i$, respectively, and the Hessian of the KL-divergence as $H$. The approximation to Equation 13 is given by

$$\theta_{k+1} = \arg\max_\theta g^\top (\theta - \theta_k)$$

$$\text{s.t.} \quad b_i^\top (\theta - \theta_k) + J_{C_i}^\beta(\pi_k) - d_i \le 0, i = 1, \ldots, m \tag{14}$$

$$\frac{1}{2}(\theta - \theta_k)^\top H (\theta - \theta_k) \le \delta.$$

---

**Algorithm 1** Worst-case Constrained Policy Optimization

---

**Input:** Initial policy $\pi_0 \in \Pi_\theta$, tolerance $\alpha$ and confidence level $\beta$
**for** $i = 0, 1, 2, dots$ **do**
    Sample trajectories $\mathcal{D}_i = \{\tau\}, \tau \sim \pi_{\theta_i}$.
    Form sample estimates $\hat{g}, \hat{b}, \hat{H}, \hat{c}$ with $\mathcal{D}_i$
    **if** If the primal problem in Equation 14 is feasible **then**
        Solve dual problem in Equation 15 to get $\lambda^*$, $\nu^*$
        Compute updated policy parameters $\theta^*$ with Equation 16.
    **else**
        Compute recovery policy parameters $\theta^*$ with

$$\theta^* = \theta_k - \sqrt{\frac{2\delta}{b^T H^{-1} b}} H^{-1} b.$$

    **end if**
    Obtain $\theta_{k+1}$ by backtracking line search to enforce satisfaction of sample estimates of constraints in Equation 13.
**end for**

---

As $H$ is always positive semi-definite, the above optimization problem can be efficiently solved in its dual form when the gradient $g$ and $b_i$ are appropriately estimated. Here, $g$ can be easily obtained by taking derivative of the objective after replacing GAE with our proposed CSAE. For estimating the gradient $b_i$ of the CVaR constraint $J_{C_i}^\beta$, we adopt the likelihood estimate proposed by Tamar et al. (2015):

$$b_i = \nabla_\theta J_{C_i}^\beta \mathbb{E}_{\substack{\tau \sim \Delta_{\pi, \beta} \\ s_0 \sim \mu}} \left[ \left( J_{C_i}^\beta(s_0) - \text{VaR}_\beta(J_{C_i}^\beta(s_0)) \right) \nabla_\theta \log \pi_\theta(a|s) \right].$$

Here $\text{VaR}_\beta(J_{C_i}^\beta(s_0))$ is empirically estimated from the batch of sampled trajectories used for each update. Then we use the same algorithm as CPO (Achiam et al., 2017) to learn the policy.

We now derive the full algorithm to solve the optimization problem in Equation 14, which can also be found in CPO Achiam et al. (2017). Let $c_i$ denotes $J_{C_i}^\beta(\pi_k) - d_i$, $B \doteq [b_1, \ldots, b_m]$ and $c \doteq [c_1, \ldots, c_m]^T$, we can express the dual to Equation 14 as follows:

$$\max_{\lambda \geq 0, \nu \succeq 0} -\frac{1}{2\lambda} \left( g^T H^{-1} B - 2r^T \nu + \nu^T S \nu \right) + \nu^T c - \frac{\lambda \delta}{2}, \tag{15}$$

where $r = g^T H^{-1} B$, $S = B^T H^{-1} B$. Solving this problem is much easier than the primal problem especially when the number of constraints is low. Let $\lambda^*, \nu^*$ denote a solution to the dual problem, the solution to the primal is given by

$$\theta^* = \theta_k + \frac{1}{\lambda^*} H^{-1}(g - B\nu^*). \tag{16}$$

We now ready to present the full algorithm in Algorithm 1.

### 7.3 EXPERIMENTAL PARAMETERS

For circle tasks, the cost function is given by

$$C(s, a, s') = \mathbf{1}[|x| > x_{\text{lim}}], \tag{17}$$

where $x$ is the horizontal position of the agent after this transition, $x_{\text{lim}}$ is a hyper-parameter specifying the location of the two vertical lines defining the safe regions.

For all the experiments, we set the discount factor $\lambda$ to be $0.995$, and the KL step size for trust region to be $0.01$. The other parameters for environment and algorithms in our experiments are listed in the following table.

| | PointCircle | AntCircle | HumannoidCircle | PointGather | AntGather |
|---|---|---|---|---|---|
| Circle Radius | 10 | 3 | 5 | - | - |
| $x_{\text{lim}}$ | 4 | 1.5 | 2 | - | - |
| Batch Size | 50k | 100k | 100 | 50k | 100k |
| Trajetory Length | 50 | 1000 | 1000 | 30 | 500 |
| Constraint Value | 5 | 10 | 10 | 3 | 1 |
| $\beta$ | 0.5 | 0.1 | 0.1 | 0.1 | 0.1 |

Table 1: Hyper-parameter settings.

### 7.4 PROOF OF $k$-STEP ADVANTAGE

We have $V(s_t) = \mathbb{E}_{a,s_{t+1}}[r_t + \gamma V(s_{t+1})]$. Rearranging it gives $\mathbb{E}_{a_t,s_{t+1}}[r_t] = V(s_t) - \gamma\mathbb{E}_{s_{t+1}}[V(s_{t+1})]$, which actually provides an unbiased estimator $\hat{r}_t$ for the expected one-step reward $\mathbb{E}_{a_t,s_{t+1}}[r_t]$, as given by

$$\hat{r}_t = V(s_t) - \gamma V(s_{t+1}). \tag{18}$$

Rewrite $\bar{r}_t$ as $\bar{r}_t = \alpha_t r_t + (1 - \alpha_t)\hat{r}_t$, then we have

$$
\begin{aligned}
A_t^{(k)} &= -V(s_t) + \bar{r}_t + \gamma\bar{r}_{t+1} + \cdots + \gamma^{k-1}\bar{r}_{t+k-1} + \gamma^k V(s_{t+k}) \\
&= -V(s_t) + \alpha_t r_t + \gamma\alpha_{t+1}r_{t+1} + \cdots + \gamma^{k-1}\alpha_{t+k-1}r_{t+k-1} + \gamma^k V(s_{t+k}) \\
&\quad + (1-\alpha_t)\hat{r}_t + \gamma(1-\alpha_{t+1})\hat{r}_{t+1} + \cdots + \gamma^{k-1}(1-\alpha_{t+k-1})\hat{r}_{t+k-1} \\
&= -V(s_t) + \alpha_t r_t + \gamma\alpha_{t+1}r_{t+1} + \cdots + \gamma^{k-1}\alpha_{t+k-1}r_{t+k-1} + \gamma^k V(s_{t+k}) \\
&\quad + (1-\alpha_t)[V(s_t) - \gamma V(s_{t+1})] \\
&\quad + \gamma(1-\alpha_{t+1})[V(s_{t+1}) - \gamma V(s_{t+2})] \\
&\quad + \cdots \\
&\quad + \gamma^{k-1}(1-\alpha_{t+k-1})[V(s_{t+k-1}) - \gamma V(s_{t+k})] \\
&= -V(s_t) + \alpha_t[r_t + \gamma V(s_{t+1}) - V(s_t)] + [V(s_t) - \gamma V(s_{t+1})] + \gamma^k V(s_{t+k}) \\
&\quad + \gamma\alpha_{t+1}[r_{t+1} + \gamma V(s_{t+2}) - V(s_{t+1})] + \gamma[V(s_{t+1}) - \gamma V(s_{t+2})] \\
&\quad + \cdots \\
&\quad + \gamma^{k-1}\alpha_{t+k-1}[r_{t+k-1} + \gamma V(s_{t+k}) - V(s_{t+k-1})] + \gamma^{k-1}[V(s_{t+k-1}) - \gamma V(s_{t+k})] \\
&= \sum_{l=0}^{k-1}\gamma^l\ \alpha_{t+l}\delta_{t+l},
\end{aligned}
$$

where $\delta_{t+l} \doteq r_{t+l} + \gamma V(s_{t+l+1}) - V(s_{t+l})$.

### 7.5 PROOF OF THEOREM 1

**Lemma 1.** *Achiam et al. (2017) For any function $f : S \to \mathbb{R}$ and any policy $\pi$,*

$$(1-\gamma)\mathbb{E}_{s\sim\mu}[f(s)] + \mathbb{E}_{\substack{s\sim d^\pi \\ a\sim\pi' \\ s'\sim P}}[\gamma f(s')] - \mathbb{E}_{s\sim d^\pi}[f(s)] = 0. \tag{19}$$

Combining this with Equation 9, we obtain the following, for any function $f$ and any policy $\pi$:

$$J_\pi^{\text{safe}} = \mathbb{E}_{s\sim\mu}[f(s)] + \frac{1}{1-\gamma}\mathbb{E}_{\substack{s\sim d^\pi \\ a\sim\pi' \\ s'\sim P}}[\bar{r}(s,a,s') + \gamma f(s') - f(s)] \tag{20}$$

In particular, we choose the function $f(s)$ to be value function $V^\pi(s)$. Thus, we have

$$J_\pi^{\text{safe}} = \mathbb{E}_{s\sim\mu}[V^\pi(s)] + \frac{1}{1-\gamma}\mathbb{E}_{\substack{s\sim d^\pi \\ a\sim\pi' \\ s'\sim P}}[\bar{r}(s,a,s') + \gamma V^\pi(s') - V^\pi(s)]$$

**Lemma 2.** *For any function:* $f : S \to \mathbb{R}$ *and any policies* $\pi$ *and* $\pi'$, *define*

$$L_{\pi,f}(\pi') \doteq \mathbb{E}_{\substack{s \sim d^\pi \\ a \sim \pi \\ s' \sim P}} \left[ \left( \frac{\pi'(a|s)}{\pi(a|s)} - 1 \right) (\bar{r}(s,a,s') + \gamma f(s') - f(s)) \right], \tag{21}$$

*and* $\epsilon_f^{\pi'} \doteq \max_s |\mathbb{E}_{a \sim \pi', s' \sim P}[\bar{r}(s,a,s') + \gamma f(s') - f(s)]|$. *Then the following bounds hold:*

$$\begin{aligned}
J^{\text{safe}}(\pi') - J^{\text{safe}}(\pi) &\geq \frac{1}{1-\gamma} \left( L_{\pi,f}(\pi') - 2\epsilon_f^{\pi'} D_{TV}(d^\pi || d^\pi) \right), \\
J^{\text{safe}}(\pi') - J^{\text{safe}}(\pi) &\leq \frac{1}{1-\gamma} \left( L_{\pi,f}(\pi') + 2\epsilon_f^{\pi'} D_{TV}(d^\pi || d^\pi) \right),
\end{aligned} \tag{22}$$

*where* $D_{TV}$ *is the total variational divergence.*

*Proof.* The proof can be established by following the one for Lemma 2 in Achiam et al. (2017) where we substitute $J_\pi^{\text{safe}}$ in Equation 20. $\square$

**Lemma 3.** *Achiam et al. (2017) The divergence between discounted furture state visitation distributions,* $\|d^{\pi'} - d^\pi\|_1$, *is bounded by an average divergence of the policies* $\pi'$ *and* $\pi$:

$$\|d^{\pi'} - d^\pi\|_1 \leq \frac{2\gamma}{1-\gamma} \mathbb{E}_{s \sim d^\pi}[D_{TV}(\pi'||\pi)[s]], \tag{23}$$

*where* $D_{TV}(\pi'||\pi)[s] = (1/2) \sum_a |\pi'(a|s) - \pi(a|s)|$.

Now, with Lemma 2 and Lemma 3, we are ready to prove Theorem 1 as follows.

*Proof.* By choosing $f(s_t) = V^\pi(s_t)$ to be the safety value function in Lemmas 2 and , we have

$$\begin{aligned}
L_{\pi,f}(\pi') &= \mathbb{E}_{\substack{s_t \sim d^\pi \\ a \sim \pi' \\ s_{t+1} \sim P}} [\bar{r}_t + \gamma V^\pi(s_{t+1}) - V^\pi(s_t)] - \mathbb{E}_{\substack{s_t \sim d^\pi \\ a_t \sim \pi \\ s_{t+1} \sim P}} [\bar{r}_t + \gamma V^\pi(s_{t+1}) - V^\pi(s_t)] \\
&= \mathbb{E}_{\substack{s_t \sim d^\pi \\ a \sim \pi' \\ s_{t+1} \sim P}} [\bar{r}_t + \gamma \bar{r}_{t+1} + \gamma^2 V^\pi(s_{t+2}) - V^\pi(s_t)] - \mathbb{E}_{\substack{s_t \sim d^\pi \\ a_t \sim \pi \\ s_{t+1} \sim P}} [\bar{r}_t + \gamma \bar{r}_{t+1} + \gamma^2 V^\pi(s_{t+2}) - V^\pi(s_t)] \\
&= \mathbb{E}_{\substack{s_t \sim d^\pi \\ a \sim \pi' \\ s_{t+1} \sim P}} [\bar{r}_t + \cdots + \gamma^{k-1} \bar{r}_{t+k-1} + \gamma^k V^\pi(s_{t+k}) - V^\pi(s_t)] \\
&\quad - \mathbb{E}_{\substack{s_t \sim d^\pi \\ a_t \sim \pi \\ s_{t+1} \sim P}} [\bar{r}_t + \cdots + \gamma^{k-1} \bar{r}_{t+k-1} + \gamma^k V^\pi(s_{t+k}) - V^\pi(s_t)] \\
&= \mathbb{E}_{\substack{s_t \sim d^\pi \\ a \sim \pi' \\ s_{t+1} \sim P}} [A_t^k] - \mathbb{E}_{\substack{s_t \sim d^\pi \\ a \sim \pi \\ s_{t+1} \sim P}} [A_t^k]
\end{aligned}$$

Thus, computing the exponentially average of $L_{\pi,f}\pi'$ with $\lambda$ as the weighting cofficient gives:

$$L_{\pi,f}(\pi') = \mathbb{E}_{\substack{s_t \sim d^\pi \\ a \sim \pi' \\ s_{t+1} \sim P}} [A_t^{\text{CSAE}}] - \mathbb{E}_{\substack{s_t \sim d^\pi \\ a \sim \pi \\ s_{t+1} \sim P}} [A_t^{\text{CSAE}}] \geq \mathbb{E}_{\substack{s_t \sim d^\pi \\ a \sim \pi' \\ s_{t+1} \sim P}} [A_t^{\text{CSAE}}]. \tag{24}$$

The last inequality comes from the fact that $\mathbb{E}_{\substack{s_t \sim d^\pi \\ a \sim \pi \\ s_{t+1} \sim P}} [A_t^{\text{CSAE}}] \leq 0$. Then applying Lemma 3 gives

the result.

$\square$

### 7.6 PROOF OF THEOREM 2

Define $\xi$ to be the $\beta$-worst-case distribution over the trajectories, *i.e.*, $\xi(\tau) = 1/\beta$ if $C(\tau)$ is among the top $\beta$ most costly trajectories; and $\xi(\tau) = 0$ otherwise. Denote $P_\beta = \xi \circ P$ to be the weighted probability distribution and $d_\beta^\pi$ to be the discounted future state distribution for the $\beta$-worst cases.

Then the expected cost over the $\beta$-worst-case trajectories can be expressed compactly as:

$$J_C^\beta(\pi) = \frac{1}{1-\gamma} \mathbb{E}_{\substack{s \sim d_\beta^\pi \\ a \sim \pi \\ s' \sim P_\beta}} [C(s, a, s')]. \tag{25}$$

We also have the following identity:

$$(I - \gamma P_\beta) d_\beta^\pi = (1 - \gamma)\mu. \tag{26}$$

With the above relation, we can obtain the following lemma.

**Lemma 4.** *For any function $f : S \to \mathbb{R}$ and any policy $\pi$,*

$$(1 - \gamma)\mathbb{E}_{s \sim \mu}[f(s)] + \mathbb{E}_{\substack{s \sim d_\beta^\pi \\ a \sim \pi \\ s' \sim P_\beta}} [\gamma f(s')] - \mathbb{E}_{s \sim d_\beta^\pi}[f(s)] = 0. \tag{27}$$

Combining with Equation 25, we have

$$J_C^\beta(\pi) = \mathbb{E}_{s \sim \mu}[f(s)] + \frac{1}{1-\gamma} \mathbb{E}_{\substack{s \sim d_\beta^\pi \\ a \sim \pi \\ s' \sim P_\beta}} [C(s, a, s') + \gamma f(s') - f(s)]. \tag{28}$$

Choosing the cost value function $V_C^\pi$ as $f$ gives:

$$J_C^\beta(\pi) = \mathbb{E}_{s \sim \mu}[V_C^\pi(s)] + \frac{1}{1-\gamma} \mathbb{E}_{\substack{s \sim d_\beta^\pi \\ a \sim \pi \\ s' \sim P_\beta}} [C(s, a, s') + \gamma V_C^\pi(s') - V_C^\pi(s)]. \tag{29}$$

Following the proof for Theorem 1, we obtain Theorem 2.

## 7.7 EXPERIMENTS ON WCMDP

To further study how our two contributions (CSAE and WCMDP) contribute to the final algorithm, we perform ablation study where the safe algorithm does not dampen the advantage function but respects the worst-case constraints, which is referred as WC in the following. We compare WC with CSAE, CSAE-WC and the other baseline methods in Fig. 5. Compared to CSAE, though WC is able to give better safety guarantee, it actually produces inferior performance, especially on PointCircle and AntGather. Besides, CSAE also demonstrates faster convergence speed than WC. By involving them in the same algorithm, CSAE-WC is able to combine their strengths and overcome their weaknesses, thus results in superior return performance and safety guarantee.

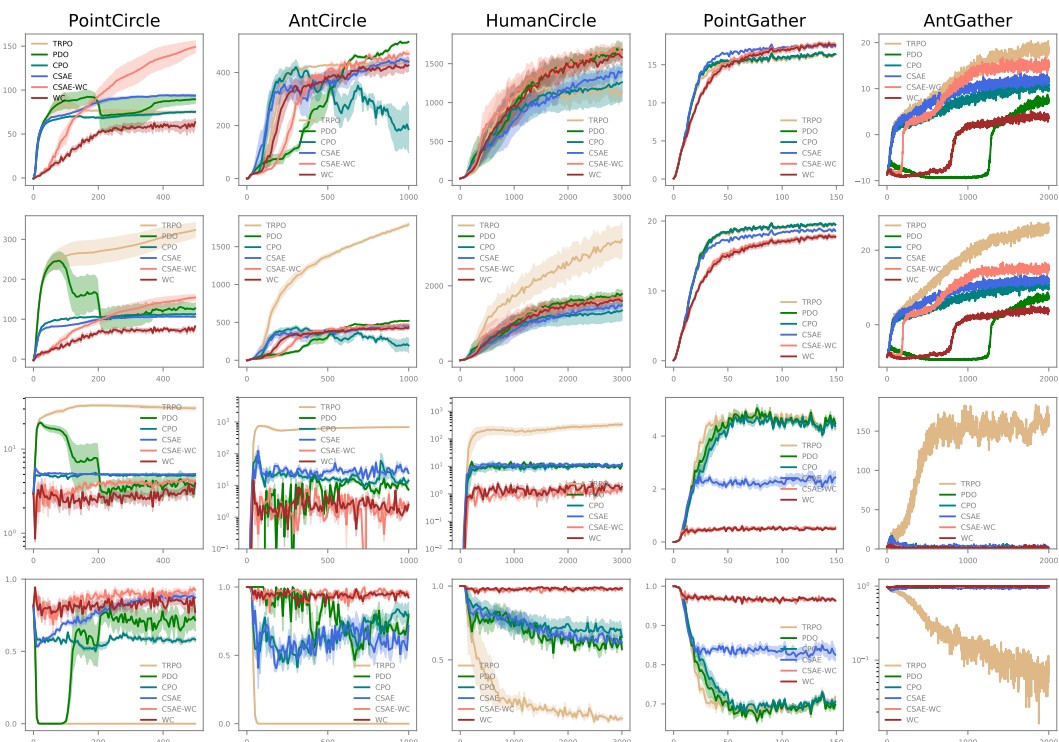

Figure 5: Learning curve comparison between our methods (CSAE, WC and CSAE-WC) and the state-of-the-arts (TRPO, PDO, CPO) for five safe RL problems. First row: safe cumulative reward. Second row: total cumulative reward. Third row: cumulative cost. Fourth row: ratio of safe trajectories.

