# OpenReview forum: "Learning Safe Policies with Cost-sensitive Advantage Estimation"
_ICLR.cc/2021/Conference — Reject_

### Official Review · AnonReviewer1 · 2020-10-25
**A worst-case (i.e. CVaR) variant of "Constrained Policy Optimization" (Achiam et al. 2017)**

**Rating:** 5
**Confidence:** 2

**Review:**

The authors propose two contributions:
1 - A Cost-Sensitive Advantage Estimation procedure called CSAE which is intended to differentiate more clearly safe and unsafe state-actions pairs with high rewards;
2 - A beta-quantile worst-case formulation of the Constrained MDP called WCMDP, and a variant of CPO  to solve this problem instead of its usual expectation formulation.
I liked the second part which prevents constraint violation with high confidence but I found the part on CSAE hard to follow.
The notion of safety being defined here in term of trajectory budget, not in term of state, I had difficulties to understand the notion of "safe-states" or "safe state action pairs"  when opposed to the notion of "safe-policies" as defined for instance in (Achiam et al. 2017). This notion of safe-sates seems however to be a key element for the definition of the CSAE estimate which is a variant of the "Generalized Advantage Estimation" (Schulman et al. 2016) where the TD error is nullified on "unsafe states".
Is it because the considered policies are stationary ?
Probably for the same reason I did not understand the claim of the authors on page 2 that "CPO sacrifices too much on the expected return".
The proposed algorithm is detailed in the Appendix, its main difference with the CPO algorithm is the formulation of the primal-dual problem of equation (14) which rely on a beta-worst case discounted cost (and CSAE) instead of the expected discounted cost.
Theorem 1 gives a TRPO-like lower bound similar to the ones in  (Schulman et al. 2016).
The experiments in section 5 are performed on the same environments as (Achiam et al. 2017), allowing for a comparison of the two methods and underlining clearly the improvement of CSAE-WC against CPO and other baselines. The improvement from CPO to CSAE seems less clear to me.
What would be the performance of a simple beta-worst case version of CPO without the CSAE machinery ?

Minor remarks:
A few typos did add to my confusion when delving into the paper, for instance on equation (2) and on page 4 when introducing the gradient CSAE the authors forgot the log in the policy gradient formula, a strange mistake that is hopefully not committed in the appendix.
The TD error delta_t should be defined before  equation (3) on page 4.

---

> ### Author Response · Authors · 2020-11-19
> **Response to AnonReviewer1: Clarification on our problem setting and ablation of WCMDP**
>
> Thank you for acknowledging our contribution in developing the worst-case MDP. For your negative comments, we believe that it is a misunderstanding resulted by our unclear presentation. We will clarify as follows.
>
> **A1.** *[notion of “safe-states” and CSAE]*
>
> Please see [our response to all reviewers](https://openreview.net/forum?id=uVnhiRaW3J&noteId=4xP6pjhFdPz) for details.
>
> **A2.** *[Experiments on beta worst-case version of CPO]*
>
> Please refer to [our response to all reviewers](https://openreview.net/forum?id=uVnhiRaW3J&noteId=4xP6pjhFdPz) for details.
>
> **A3.** *[Minor remarks]*
>
> We will fix the typos accordingly, thank you for pointing out.

---

### Official Review · AnonReviewer4 · 2020-10-28
**Review of "Learning Safe Policies with Cost-sensitive Advantage Estimation"**

**Rating:** 7
**Confidence:** 4

**Review:**

Summary: The authors propose to improve a safe RL algorithm, constrained policy optimizaiton, that can learn the optimal safe policy while exploring unsafe states less often during the training process. In particular, they dampen the estimated advantage associated with unsafe states, which encourages the RL algorithm to explore safe states more often during the learning process. In addition, the authors aim to find a policy that satisfies the constraints with high probability, rather than only in expectation, by considering the worst-case constraints. The empirical results show that a safe RL algo that dampens the advantage and respects worst-case constraints are able to learn policies with large returns and avoid unsafe states.

Pros:
+ The authors present a simple modification of a safe RL method to make it learn faster and more safely. The empirical results validate this claim.

Major concerns:

1. The authors describe how their method replaces the reward for an unsafe state as the average reward, and claim that this is generally better than previous reward reshaping methods, e.g. setting the reward for unsafe states to zero. The authors only present a single example that compares the two approaches (Fig 4); However, it's not clear if the average reward is a wider range of scenarios. I'd like to see a more comprehensive comparison in Fig 1 to justify this claim. Also, can the authors give an explanation for why the mean is a more suitable penalization of unsafe states or is more universally applicable, compared to other reward reshaping methods?

2. The authors compare the cumulative safe reward of different RL algorithms. However, the theoretical results in Theorem 1 compare the cumulative reshaped rewards. It is somewhat difficult to interpret the result in Theorem 1 since the reshaped reward uses the average reward for unsafe states, rather than setting the reward to zero. Given the similarity between there two rewards, can the authors establish a bound on the cumulative safe reward instead, as this seems to be the quantity we are truly interested in?

3. In the empirical analysis, the improvement from CSAE to CSAE-WC was quite large, whereas the improvement from CPO to CSAE was smaller. To understand how much dampening the advantage improves on the worst-case constraints, can the authors do an ablation study where the safe RL algo does not dampen the advantage function but respects the worst case constraints?

Minor concerns:

1. Please plot the constraint threshold in Fig 1 row 2. It's hard to tell if the methods have satisfied the constraints, since there is no reference line.
2. There is a typo when referring to the rows in Fig 1 -- see last paragraph on page 6. It should say "third and fourth rows", rather than "second and third rows".

---

> ### Author Response · Authors · 2020-11-19
> **Response to AnonReviewer4: adding required experiments**
>
> **A1.** *[Theorem 1??]*
>
> The cumulative reward guarantee from the safe states can be directly obtained from Theorem 1. By Eqn. (18), we have $\hat{r}_t = V(s_t) - \gamma (V_{s+1})$.
>
> Substituting this into the proof of Theorem 1 after Eqn. (23), we can obtain the first term within the definition of $L_{\pi, f}(\pi)$ is
>
> $E[\bar{r_t} + \gamma V^\pi (s_{t+1}) - V^\pi (s_t)]  = E[\alpha_t*{r_t} + \alpha_t* \gamma V^\pi (s_{t+1}) - \alpha_t* V^\pi (s_t)]$
>
> which is corresponding to the cumulative reward from the safe states (whose $\alpha_t=1$).
>
> **A2.** *[More experiments on W-CMDP]*
>
> Please refer to our [response to all reviewers](https://openreview.net/forum?id=uVnhiRaW3J&noteId=4xP6pjhFdPz) for details.
>
>
> **A3.** *[More experiments on other reward reshaping methods]*
>
> We would like to emphasize that we present the reward reshaping as an intuitive explanation for our CSAE method, which is actually dampening the advantage value. Designing other non-trivial reward reshaping methods is out of the scope of our work and thus we do not directly compare with them in the experiments.
>
> **A4.** *[Minor concerns]*
>
> We will revise Fig.1 accordingly, thank you for the suggestions.

---

### Official Review · AnonReviewer3 · 2020-10-28
**review for "Learning Safe Policies with Cost-sensitive Advantage Estimation"**

**Rating:** 6
**Confidence:** 3

**Review:**

The authors' rebuttal has addressed some of my confusion regarding the paper, which is greatly appreciated. The additional baseline of early termination would still be interesting to have, though I agree it's not critical for the presented line of work. In general, I think the work is interesting and will keep my current score (6).

=================================

The paper introduces a learning algorithm for training a control policy to complete certain tasks while satisfying some safety constraints. The main ideas in this work are to first use a cost-sensitive advantage function, where the advantage values for the unsafe states are set to zero. Second, a more conservative estimation of the safety cost is proposed to further improve the safety of the robot during the learning process. The authors demonstrated that with the proposed modified reward function, the algorithm would obtain a controller that completes the task while being safe. The proposed algorithm is evaluated on a set of simulated control problems and with both proposed components used, the algorithm achieves better performance in terms of both task completion and safety than prior methods.

The paper solves an important problem of training a performant control policy while taking the safety of the robot into consideration. The paper is well written and the experiments show good results compared to prior methods.

However, I do have a few questions about the paper:
1. The paper mentioned first about zeroing the advantage for the unsafe states and showed the equivalence of doing that to a shaped reward which replaces the original reward with the shaped reward. I'm wondering if this equivalence also holds for the safe states? For example, within a trajectory that ends in an unsafe state, a prior safe state's advantage will take the original reward of the unsafe state into consideration if only the unsafe state's advantage is modified, which might be different from the shaped reward version. It would be great if the authors could help me understand this part better.
2. In the implementation of the work, is it using the modified advantage function in Eq. 4, or the shaped reward in Eq. 5? It's a bit confusing in that the title of the paper suggests the use of Eq. 4, while section 4.2 seems to be saying it's using the shaped reward? If it's using the shaped reward version, is there a reason not to use Eq. 4 directly?
3. It seems a simple baseline to compare to is to terminate the rollout when the robot enters an unsafe state, as is done in various OpenAI Gym tasks. There seems to be some analogy in this strategy and the proposed one in that when the rollout is terminated, the policy will not have any updates regarding the unsafe states, i.e. zero gradient, and thus corresponds to zero advantage in policy gradient formulation. I do note that there are still many differences between the two methods, but I feel it's worth trying given the simplicity of the approach and how it's been effective in teaching the robots to run upright in existing problems.

Overall, I think it's an interesting approach with good results.

---

> ### Author Response · Authors · 2020-11-19
> **Response to AnonReviewer3: clarification on you questions**
>
> **A1.** *[The equivalence between zeroing usafe advantage value and reshaping the reward]*
>
> Recall that our CSAE is developed based on GAE, which is given by the discounted cumulative summation of one-step TD errors (advantages).  By zeroing the advantage for unsafe states, we mean zeroing the one-step TD errors. When we calculate the actual advantage of a state (either safe or not), we are actually applying Eqn.(4) on the revised one-step TD errors, instead of directly on the original reward signals. Therefore, the equivalence still holds for safe states. See Section 7.4 in the appendix for a proof.
>
> **A2.** *[Implementation of CSAE]*
>
> CSAE is implemented using the modified advantage function is Eqn.(4). The reward reshaping is provided as an interpretation of advantage modification.
>
> **A3..** *[Terminating the rollout when the robot enters an unsafe state.]*
>
> We agree that this is an interesting idea but this deviates from our current setting that we adopted from existing works. In this work, we are considering learning safe policies in the sense that the cumulative cost of the trajectory will not exceed the allowed upper limit, instead of trying to avoid every unsafe state-actions. Thus, experiencing the unsafe states and the following states will help learn the policy giving bounded trajectory-level cost. Simply terminating the rollout however will introduce loss of some useful information from the unsafe state-actions.  For example,  the agents may not be able to learn to avoid critical states, as they would never see them during training.

---

### Official Review · AnonReviewer2 · 2020-10-30
**Recommendation to Reject**

**Rating:** 4
**Confidence:** 4

**Review:**



#### Summary:

In this work, the authors extend the CPO (Achiam et al) with a Cost-Sensitive Advantage estimation technique that eliminates the negative effect of unsafe high-reward state-action transitions. The authors then modify the CMDP objective with a CVAR based objective function and provide empirical results on the Mujoco based safety environments.


#### Strengths:

- The idea of incorporating the notion of including safe/unsafe states to build a new advantage estimation is an interesting one, and I believe the link to Cost-Sensitive Advantage Estimation and reward shaping is novel.

- The visualizations of the trajectory of the final policy returned by the algorithm were insightful to see.

#### Weakness:

- **Unclear setting:** The authors start with the motivation with the setting of the Constrained MDPs, but through-out the draft the discussion is carried in terms of safe and unsafe states. Note that, while the safe-state formulation can be defined using the CMDP framework, the opposite is not always true.


- **Technical flaws:** The method relies on an important assumption that is not discussed in enough detail. The paper assumes that the agent has knowledge about what states are safe and which states are unsafe ($\alpha_t$ variable in Eq 4). Note that, this is an important distinction, as for CMDPs defining what states are unsafe (or what states to avoid) is a big challenge in itself (Altman, 1999).
 For the purpose of this work, the authors use a hard-coded safety distinction ($\apha_t = \mathbb{1}[C(s_t,a_t,s_{t+1})] > 0 ) and though this works for the examples chosen in this work (the examples based on CPO) there is no reason this will be true in general. In most cases, an optimal policy of CMDP is the stochastic policy, and directly avoiding some states is not always possible, and usually, the safe or unsafe states for CMDP are a function of the policy.

 This is also related to the author's claim that their proposed method is "free of hyper-parameter tuning and easy to deploy". In this case, the authors have already abstracted the difficult aspect of finding safe and unsafe states by including that information in the $\alpha$ variable, and the authors assume that they always have access to this variable. As I mentioned above this is not always true. Furthermore, if the agent already knows safe and unsafe state distinction, there are much simpler and computationally efficient methods (compared to CPO) are available that can be deployed, for instance, (Dalal et al, 2018) that the authors mention in the manuscript.

- **Incremental nature of work**: The proposed method relies heavily on the CPO (Achiam et al), both for the theoretical and empirical results and baselines. Combined with the above flaw of hard-coding the unsafe states, it is not clear what is the novelty of this work. The Worst-Case formulation uses a CVAR based objective instead of expectation based. While that is an interesting formulation, there are no comments made on how that changes the quality of the solution compared to the regular objective. Are the optimal policy in regular CMDP and CVAR-based objective different? How tough the problem becomes?

- **Reproducibility**: There is no mention of code release for empirically intense work.

- **Baselines:** The authors mention that CPO is the SOTA, but that is not true. From (Ray et al, 2019), we know that Lagrange based approaches can actually perform better than CPO when trained properly.

#### References:

- Ray, Alex, Joshua Achiam, and Dario Amodei. "Benchmarking safe exploration in deep reinforcement learning." arXiv preprint arXiv:1910.01708 (2019).

---

> ### Author Response · Authors · 2020-11-19
> **Response to AnonReviewer2: DO NOT agree there are technical flaws**
>
> **A1.** *[Unclear setting]*
>
> As we explained in the introduction (paragraph 3 and 4),  CPO sacrifices too much of the
> expected return for learning the safety policy. To overcome this problem, we propose CSAE to gain more reward by better estimating the advantage value. We follow exactly the same setting/assumptions as CPO, safe and unsafe states are only used in CSAE. Please see A2 for more explanation.
>
> **A2.** *[Technical flaws]*
>
> This is not a technical flaw, but a misunderstanding due to our unclear presentation. The reviewer misunderstood the alpha as a provided signal but this is not true. Alpha is defined from the received cost during the agent training which is a standard setting. Please see our [reply to all reviewers](https://openreview.net/forum?id=uVnhiRaW3J&noteId=4xP6pjhFdPz) for details.
>
> **A3.** *[Incremental nature of work]*
>
> We respectfully disagree with the comment saying that our paper is incremental. Our contributions are two-fold: the CSAE algorithm and the WCMDP model. The CSAE algorithm offers a new advantage estimator for solving safe RL problems. The novelty of the link between CSAE and reward reshaping is also acknowledged by the reviewers. Though our derivation of the theoretical guarantees for CSAE is based on CPO, the proof technique is not trivial. Besides,  establishing the baseline based on CPO makes our method  directly comparable with CPO. We do not agree such common practice would make the work incremental.  The novelty of WCMDP (a CVaR version of CMDP) is also acknowledged by AR5 “The CVaR version of CMDP is also a novel and interesting setting.”
>
> *[Are the optimal policy in regular CMDP and CVAR-based objective different? How tough the problem becomes?]*
>
> CVaR-based objective provides a stronger safety guarantee with less cost-constraint violation, as it provides guarantees on the $\beta$-worst trajectories, which cannot be guaranteed by CMDP. This has been explained by Theorem 2 in the appendix. Besides, using the CVaR-based objective requires a new optimization algorithm, as detailed in the text before Eqn. (15) in the appendix.
>
> **A4.** *[Reproducibility]*
>
> Our implementation is based on CPO, and can be achieved by slightly modifying the code of CPO. We will give a detailed instruction for the modification.
>
> For CSAE, the implementation can be achieved by inserting the following code into https://github.com/jachiam/cpo/blob/master/algos/safe/sampler_safe.py#L215
> ```
> deltas *= path['safety_rewards'] == 0
> ```
>
> For WCMDP, we first obtain the corresponding data for optimization by inserting the following code into https://github.com/jachiam/cpo/blob/master/algos/safe/sampler_safe.py#L382
>
> ```
>                 if self.algo.safety_worst_ratio < 1.0:
>                     tmp = [path['safety_returns'][0] for path in paths]
>
>                     def get_safety_inds(ratio):
>                         sorted_inds = np.argsort(tmp)[::-1]
>                         safety_inds = np.sort(sorted_inds[:int(ratio * len(paths))])
>                         return safety_inds
>
>                     safety_inds = get_safety_inds(self.algo.safety_worst_ratio)
>
>                     samples_data['safety_inds'] = safety_inds
>                     samples_data['worst_safety_eval'] = 0 if len(safety_inds) == 0 else np.mean([paths[i]['safety_returns'][0] for i in safety_inds])
>                     samples_data['worst_safety_rescale'] = 1.0
>                     samples_data['worst_observations'] = tensor_utils.concat_tensor_list([paths[i]["observations"] for i in safety_inds])
>                     samples_data['worst_actions'] = tensor_utils.concat_tensor_list([paths[i]["actions"] for i in safety_inds])
>                     samples_data['worst_advantages'] = tensor_utils.concat_tensor_list([paths[i]["advantages"] for i in safety_inds])
>                     samples_data['worst_weights'] = tensor_utils.concat_tensor_list([paths[i]["weights"] for i in safety_inds])
>                     samples_data['worst_agent_infos'] = tensor_utils.concat_tensor_dict_list([paths[i]["agent_infos"] for i in safety_inds])
>                     samples_data['worst_safety_values'] = worst_safety_vals = tensor_utils.concat_tensor_list([paths[i][safety_key] for i in safety_inds])
>
> ```
> We will give a complete implementation later.
>
> **A5.** *[baselines]*
>
> The primal-dual optimization (PDO) baseline (See Sec.5 and the Sec.7 of [1] for details) in our experiments is indeed the Lagrange based methods (based on TRPO), which are shown inferior to our methods. See the experimental section for more details.
>
> [1] Achiam, Joshua, et al. "Constrained policy optimization." arXiv preprint arXiv:1705.10528 (2017).

---

### Official Review · AnonReviewer5 · 2020-11-14
**New constrained RL algorithm. Setting is unclear, and the two constributions seem disconnected.**

**Rating:** 5
**Confidence:** 3

**Review:**

Summary
In this paper, the authors proposed a new constrained policy optimization algorithm and a worst-case version of the constrained MDP framework. Tho proposed constrained policy optimization algorithm is based on CPO, and a novel advantage function (CSAE) based on the concept of a "safe" state. Experiments in control simulation tasks are provided.

In general, the main novel contribution of this work is the cost-sensitive advantage estimation (CSAE), which is later justified by the empirical study. However, I think the problem setting needs further clarification, and the new advantage estimation and reward shaping are not justified enough.

Pros:
1. The connection between the CSAE with a new form of reward is interesting. I think it provides more intuition or a different understanding of the advantage function.

2. The CVaR version of CMDP is also a novel and interesting setting.

Cons:
1. It is not clear enough what is the problem formulation that this work (or CSAE algorithm) aims to solve. Before section 4 it seems that the goal is to maximize the reward while keeping the *trajectory-level* cost under a budget. However, later the CSAE algorithm leverages the concept of "safety" of states. While the trajectory-level cost may or may not be decomposed into some state-level binary constrained, it seems either the algorithm design needs more justification or the problem setting is misleading.
Furthermore, I think the choice of $\alpha_t$ in the algorithm needs to be discussed extensively: instead of saying "e.g. C_t > 0", it needs to discuss how to choose the threshold, how the state-level constrained relate to the constraints on the discounted sum, etc.

2. Since the new advantage function is linked to the new reward form in Eq (5), it needs more discussion about why this form of reward is a natural one. It isn't obvious why using the average reward for the "unsafe" state-action pairs can penalize them.

3. The paper proposed two contributions but I did not follow how they are connected to each other. WCMDP can be applied to any constrained policy optimization algorithm.

---

> ### Author Response · Authors · 2020-11-19
> **Response to AnonReviewer5: clarification for problem formulation and algorithm design**
>
> **A1.** *[The problem formulation is not clear.]*
>
> Please see the response to all reviewers for clarification on $\alpha_t$.
> We here would like to clarify the function of CSAE which may be misunderstood by the reviewer.
> The goal of our work is indeed “to maximize the reward while keeping the trajectory-level cost under a budget”, as the reviewer mentioned. To this end, we propose CSAE to gain more reward and use the worst-case CMDP to ensure the trajectory-level cumulative cost does not exceed the constraint. We will clarify as follows.
>
> We agree that “the trajectory-level cost may or may not be decomposed into some state-level binary constrained”. However, as we explained above, we are only binarizing the state-level cost for generating pseudo labels for CSAE, our algorithm do not directly minimize the binarized state-level safety signal. CSAE is only used to estimate the advantage value of a trajectory during policy optimization after it is collected from the environment.
>
> **A2.** *[why using the average reward for the "unsafe" state-action pairs can penalize them.]*
>
> As discussed above, the CSAE is mainly used to gain more rewards by estimating the advantage value better.  This is derived from zeroing the advantage value of “unsafe” state-action pairs as explained in Sec. 4.1 of the submission. The reward reshaping (replace the reward for unsafe state-actions with the average) provides another interpretation for understanding effects of the advantage zeroing in CSAE. Intuitively, we are aiming to penalize the unsafe state-actions giving high rewards.  Using the average reward (instead of the actual high reward) will penalize the unsafe state-actions and encourage the agent to gain more rewards from safer ones.
>
>
> **A3.** *[How CSAE and WCMDP are connected?.]*
>
> As the reviewer mentioned,  the goal of this work is “to maximize the reward while keeping the trajectory-level cost under a budget”. To this end, we propose CSAE to gain more reward and the worst-case CMDP to ensure the trajectory-level cost to be under constrained, which has been discussed in the introduction and Section 4 of our paper.

---

### Author Response · Authors · 2020-11-19
**To all reviewers: clarification on the notion of state safety and ablation of WCMDP**

**1. Clarification on the notion of state safety ($\alpha$) used in CSAE**

We notice that most of the negative comments are from **misunderstanding $\alpha$ defined in CSAE due to our unclear presentation**. We will clarify as follows.

The goal we propose CASE is to gain more reward in cost-sensitive environments by better estimating the advantage values. As pointed out by the reviewers, the calculation of CSAE (Eqn.(4)) relies on the safety variable $\alpha$. We would like to clarify that $\alpha$ is NOT “proposed as a perfect metric measuring safety of a state” and does NOT indicate whether a state should be avoided or not. Instead, $\alpha$ is defined to record the experienced cost during training for developing CASE.

More specifically, we do NOT assume the agent knows which states are safe or not. Instead, we **only assume that the agent can receive that cost value caused by per transition during interacting with the environment**. This is a common assumption in most safe RL papers (Dalal et al, 2018, Achiam et al). Based on the received cost signal, we define the variable $\alpha$ (in Eq.4) as $\alpha_t = \mathbb{1}[C(s_t,a_t,s_{t+1})] > 0$ for each state in the collected trajectories, which **can be interpreted as pseudo labels generated for the transitions within the collected trajectories indicating the cost level of states**. The $\alpha$ variable is only applied when a trajectory is collected from the environment to obtain a better advantage estimation (for encouraging the agents to collect reward from less-costly states) as explained in Sec.4.1.  There might be different ways to design $\alpha$, but we empirically find our binarization choice works well. We leave exploring other forms of $\alpha$ as future work. We will make this clear in the next version.

**2. Ablation study on worst-case constrained safe RL**

We obtain a variant (referred as WC) of CSAE-WC by removing the CSAE and only applying the worse-case constraints. We added extensive experiments to study its performance in five cost-sensitive experiments. The results show that WC frequently gives performance degradation compared with CSAE-WC, the full algorithm, though it is effective at enhancing safety guarantee. In contrast, CSAE-WC is able to combine the benefits of CSAE and WC, and overcome their limitations. For the training curves and detailed explanation, please refer to Sec. 7.7 in the appendix of our revised draft.

---

### Decision · Program_Chairs · 2021-01-07
**Final Decision**

**Decision:**

Reject

**Comment:**

The reviewers appreciate the importance of enforcing safety in RL, and the technical directions considered in the paper related to incorporating cost in advantage estimation.  However, they express several concerns about the formulation of the problem considered and the consistency of the approach, as well as the somewhat incremental contribution w.r.t. CPO.  Three reviewers recommend rejection.